# Transitioning to Combined EBUS EUS-B FNA for Experienced EBUS Bronchoscopist

**DOI:** 10.3390/diagnostics11061021

**Published:** 2021-06-02

**Authors:** Jeffrey Ng, Hiang Ping Chan, Adrian Kee, Kay Leong Khoo, Kay Choong See

**Affiliations:** 1Division of Respiratory and Critical Care Medicine, University Medicine Cluster, National University Health System, 1E Kent Ridge Road, NUHS Tower Block Level 10, Singapore 119228, Singapore; hiang_ping_chan@nuhs.edu.sg (H.P.C.); adrian_cl_kee@nuhs.edu.sg (A.K.); kay_leong_khoo@nuhs.edu.sg (K.L.K.); kay_choong_see@nuhs.edu.sg (K.C.S.); 2Department of Medicine, Yong Loo Lin School of Medicine, National University Singapore, 10 Medical Drive, Singapore 117597, Singapore

**Keywords:** endobronchial ultrasound, endoscopic ultrasound, lung cancer, mediastinal lymphadenopathy and training endoscopic ultrasound

## Abstract

Endobronchial ultrasound (EBUS) combined with trans-esophageal endoscopic ultrasound bronchoscope guided fine need aspirate (EUS-B FNA) of mediastinal lymph nodes is an established procedure for diagnosis. The main barrier to a combined EBUS EUS-B FNA approach is availability of trained and accredited pulmonologist who can perform procedure safely and confidently. To address this gap, we undertook a training program for experienced EBUS bronchoscopists to train, learn, and incorporate combined EBUS EUS-B FNA into their procedural practice. Thirty-two patients were selected based on CT and or PET findings. Four experienced bronchoscopists participated by reading through learning material, observing 5 cases before performing EUS-B FNA under direct supervision. Forty-one lymph nodes and 6 non-nodal lesions were sampled. EUSAT assessment was performed by supervisor. Learning curves were derived from assessment scores. We observed that learning curve tends to plateau when participant can perform 3 or more consecutive cases with EUSAT score above 50. There were no complications. Our experience suggests that there is relative ease in transition to combined EBUS EUS-B TBNA procedures for mediastinal lymphadenopathy and lung cancer diagnosis and staging for experienced bronchoscopist using a program which incorporates direct supervision, EUSAT assessment, and extension of EUS B FNA training into daily real-world practice.

## 1. Introduction

Endobronchial ultrasound (EBUS) combined with trans-esophageal endoscopic ultrasound bronchoscope guided fine need aspirate (EUS B FNA) of mediastinal lymph nodes is an established procedure for diagnosis of mediastinal lymphadenopathy [1], lung cancer diagnosis [2], staging [3,4], and molecular testing [5]. In a meta-analysis [6], diagnosis of mediastinal lymphadenopathy combining EBUS TBNA (transbronchial needle aspirate) with EUS-B FNA gained additional positive diagnostic yield of 7.1%. In lung cancer nodal staging, combined approach increased sensitivity by 12%, translating to numbers needed to test of 25, and, in the same meta-analysis [7], authors found no difference between use of bronchoscope versus conventional echoendoscope in EUS FNA. EUS-B allowed access to inferior mediastinal lymph nodes and para-esophageal masses that are not accessible by trans-tracheal EBUS TBNA. Other reported advantages of EUS-B present when patients have excessive cough, desaturation due to poor respiratory reserve, raised intra cranial pressure, and poor visualization of lymph node target. Patient centric benefits include lower cost, single setting, proceduralist, and scope use. Complication rate is low.

There are currently no training guidelines for pulmonologists performing EUS-B FNA. Experienced EBUS TBNA bronchoscopists [8] have performed safe and accurate EUS-B FNA procedures using results from cusum analysis to demonstrate competency. The suggestion is that pulmonologist experienced in EBUS TBNA can transit to EUS-B FNA with relative ease.

The main barrier to combined EBUS EUS-B FNA approach is availability of a trained and accredited pulmonologist who can perform procedure safely and confidently. To address this gap, we undertook a training program for experienced EBUS bronchoscopists, defined as having done at least 40 EBUS TBNA a year and having 3 or more years of prior experience in EBUS TBNA, to train, learn, and incorporate combined EBUS EUS-B into their procedural practice.

## 2. Materials and Methods

### 2.1. Study Design

From 2018 to 2021, patients with mediastinal lymphadenopathy or suspected lung cancer assessed by CT or PET scan to have lesions accessible by EUS-B and difficult bronchoscopy were selected. EUS-B FNA was done on selected patients if para-esophageal lesions were inaccessible by EBUS or if there were patient factors, such as excessive cough prohibiting proper visualization, desaturation, or raised intra cranial pressure.

Four experienced EBUS bronchoscopists, with prior knowledge of lung cancer staging, EBUS scope handling and interpretation of endoscopic images of mediastinal structures, were included in the training program. Learning material on technique and anatomical landmarks of EUS-B were distributed to the four participants [9]. They observed 5 cases each performed by author (J.N.) who gained accreditation after overseas hands-on-training and participation in more than 100 cases. The trainees then went on to perform EUS-B FNA under direct supervision by author (J.N.). Verbal advice and taking over of procedure after 3 attempts were administered at supervisor’s discretion. Responsibility of patient safety, equipment, quality of specimen, and diagnostic yield was undertaken by supervisor. All bronchoscopists agreed to participate in this training audit. Institutional ethics board application number: 2021/00425.

### 2.2. Standardized EBUS FNA Procedure

Patients received topical 1% lidocaine spray for oropharyngeal anesthesia. They were placed in supine position and received conscious sedation with intravenous fentanyl and midazolam. Airway inspection using flexible bronchoscope is routinely done. Aliquots of 1% lignocaine are administered through the flexible bronchoscope for airway anesthesia.

All EBUS procedures were performed with a flexible convex probe ultrasound bronchoscope with linear scanning transducer 7.5 MHz (CP-EBUS, BF-UC260FW, Olympus, Tokyo, Japan). Two types of EBUS needle were available for use (1) 22-G needle (NA-201SX-4022, Olympus, Tokyo, Japan) (2) 25G or 22G needle (Expect, Boston Scientific, Natick, MA, USA).

Cases that required EUS-B FNA after EBUS assessment or were assessed to require only EUS-B FNA had standardized EUS-B procedure as described previously [3,4]. EUS-B was introduced into esophagus by mouth and oropharynx. In accordance with EUSAT assessment method described by Konge et al. [10], the order of identification of landmark is as follows: liver, adrenals, coeliac axis, station 7, and then station 4 left. Needle sheath is pushed out and visualized before puncturing the lymph node under ultrasonic visualization. After removal of stylet, multiple needle aspirations are made. Needle is fully retracted before removal from EBUS scope for specimen collection. To prevent upstaging of lung cancer, all FNA biopsies were done in following order of M1 distant metastases followed by N3, N2, and N1. Ultrasound images of lymph node stations and fine needle aspirate of lymph nodes are recorded and kept in patient electronic records. Close attention and supervision are given to needle handling for assurance of patient and equipment safety. Figure 1. provides CT and endoscopic ultrasound images for illustration of EUS-B FNA.

Rapid on-site cytology evaluation (ROSE) was done routinely. Aspirated material was smeared onto glass glides, air dried, and fixed in 95% ethanol. Air-dried smears were subsequently stained using Hemocolor stain (Merck, Burlington, MA, USA) and were examined by on-site cytotechnician, who would categorize samples as adequate, defined as more than 40 lymphocytes in high power field or presence of clusters of anthracotic pigment-laden macrophages, or inadequate. Excess material was placed in 10% formalin or directly brought to the laboratory for embedding into paraffin cell blocks for histologic examination using hematoxylin and eosin (H&E) staining. Immunohistochemical staining was done as required to further delineate the type of malignancy.

### 2.3. Assessment

From point of performing EUS-B FNA under direct supervision, participants were assessed prospectively by EUSAT assessment tool which was found to be valid and reliable for assessing EUS FNA performed by training pulmonologist with an echoendoscope. EUSAT has twelve items with a scale of 1–5, with a maximum of 60 points. In our assessment done on site or shortly after completion of procedure, 5 points were assigned for performance of itemized skill on first attempt, 4 points for performance at 2nd attempt with verbal guidance, 3 points for performance after more than 2 attempts, 2 points for performance after more than 2 attempts with verbal guidance, and 1 point for inability to perform itemized skill, and scope is taken over by supervisor.

### 2.4. Data Collection and Outcomes

Patient characteristics, participant profile, number of lymph nodes sampled, number of passes made per lymph node, size of lymph node, and whether EUS B FNA result was consistent with final diagnosis was collected. Final diagnosis is based on histological reports, surgical biopsy if available, or electronic records at 6-month follow up.

### 2.5. Analysis

EUSAT assessment scores were collated and plotted to display individual learning trajectory curves. Statistical analyses using Fisher exact test and logistic regression were performed to investigate association between EUSAT score and correct diagnosis, as well as between EUSAT score, lymph node size, and location. Lymph node location was dichotomized to commonly assessed stations (i.e., 4L and 7) and less frequently assessed stations (i.e., non-4L/7). Statistical analyses were performed using Stata (version 15, Statacorp, TX, USA). Statistical significance was taken to be *p* < 0.05.

## 3. Results

Characteristics of patients are as presented in Table 1: 32 patients, 27 males and 5 females, with pre procedural diagnosis ranging from malignancy to benign conditions had combined EBUS EUS-B FNA done for investigation. Five patients had further procedures done after EBUS EUS-B FNA. Twenty-seven patients had 6 or more months of follow up and EUS-B FNA result was consistent with final diagnosis in 28 patients. In 4 patients where EUS-B FNA was inconsistent with final diagnosis, 1 was diagnosed on EBUS TBNA, 1 case had radiological progression, and 2 cases were diagnosed on EBUS TBNA of lung mass lesion.

Participant characteristics are as in Table 2: four male pulmonologists, with an average of 18-years post-graduation, 10 years post specialization, and performance of 42.5 EBUS TBNA procedures per year.

Sampled lymph node characteristics are in Table 3. EUS-B FNA was done on total of 47 lesions, 41 nodal and 6 non-nodal: One station 8; and 40 station 7 and 4L lymph nodes. Four para-esophageal lung mass lesions and 2 enlarged left adrenal glands were sampled. Two and a half passes were performed per LN. Average size of lymph node lesions sampled is 20.8 mm by 14.3 mm.

No complications, such as mediastinitis or pneumothorax, were reported. In EUS-B FNA procedures, there were no cases of hypoxaemia and bleeding requiring administration of hemostatic agents.

EUSAT scores are presented in Figure 2. There were 32 sets of EUSAT score assessment done. Scores range from 37 to 55. Statistical analysis using logistic regression found no association between lymph node short axis diameter and non-4L and 7 target locations with high EUSAT score, defined as greater than 50, either on univariate or multivariate analysis, as in Table 4. No association between correct EUS-B FNA diagnosis with high EUSAT score was found (Fisher exact test, *p*= 0.109).

## 4. Discussion

We have found relative ease in training and transitioning experienced EBUS bronchoscopists to EUS-B FNA. Based on learning trajectory curves that tend to plateau at EUSAT score of 50 and above, we propose that experienced bronchoscopists achieving a supervisor assessed score 50 or above for 3 or more cases could be competent to perform EUS-B FNA independently. Using logistic regression, we did not find any association between lymph node short axis diameter and location, or even 4L/7 target location, with high EUSAT score, defined as more than 50. We suggest that temporal improvement in EUSAT scores is attributable to learning effect over time. No association between correct diagnosis and high EUSAT score was found on analysis, and this can be accounted for by effect of direct supervision while participants are learning.

Expert guidelines for EUS FNA of pancreas recommended at least 150 cases before competency [11]. In training programs for pulmonologists to acquire EUS FNA skills using echoendoscope, trainees may not have sufficient competency even after 20 cases [12]. EUS FNA of mediastinal lymph nodes is reckoned to be technically less difficult than EUS FNA of pancreas [13]. Endo-bronchoscope is easier to handle than esophagogastroscope, hence the relative ease and fewer number of cases required to attain high EUSAT scores, in our experience.

In the study by Konge et al. [12], using EUSAT assessment tool with maximum score of 48, 20 supervised procedures were deemed to be insufficient for EUS FNA by pulmonologist using echo-endoscope. Mean score achieved by 4 pulmonologists was 35, and mean score by experienced proceduralist was 40.6. A direct comparison of our study to Konge et al. [12] is unlikely to be valid due to differences in definition of score in EUSAT tool and in use of endo-bronchoscope in our study versus echo-endoscope. Our study did not use score of 0. We allowed score of 5 to be given if itemized skill was performed successfully by trainee in 1 attempt. Use of movable needle sheath was assessed in all attempts made by trainees.

Currently, there are no established local or regional training guidelines for EUS B FNA training and accreditation. Experienced pulmonologists who wish to incorporate EUS-B into their practice will be advised to seek training through supervision by trained respiratory or gastroenterology colleague. Our experience suggests that overlapping skills acquired from respiratory and critical care practice, including handling of EBUS bronchoscope, ultrasound image interpretation, and needle puncture and aspiration, contribute to ease of transition to combined EBUS EUS-B FNA.

In the study by Leong et al. [8], learning curves for EUS-B FNA by 3 experienced EBUS bronchoscopists were described. Learning curves were generated from cusum analysis and their conclusion was that experienced EBUS bronchoscopists can perform EUS-B FNA of nodal and non-nodal lesions safely and accurately. Even though different in methodology, their conclusion and experience is consistent with ours of ease with transitioning to EUS-B FNA by pulmonologist experienced in EBUS TBNA.

Our study has limitations. Due to small number of participants, we were unable to examine the correlation between the number of previous EBUS performed and EUSAT score or the increase in EUSAT by repeat examinations. Potential source of bias from use of 1 supervisor assessor instead of multiple assessors. Waiting for more trained expert supervisors, sending video records for independent assessment, and obtaining EUSAT scores from expert performed EUS-B FNA to have a reference score would have delayed development of EBUS EUS-B clinical services. We did not use diagnostic yield as competency outcome as the supervisor is likely to influence diagnostic yield outcome in supervised training. We are unable to account for supervisor influences and effect on EUSAT scores and participants competency to perform independently after 2-year study period. However, we have observed that our bronchoscopists could perform combined EBUS EUS-B FNA procedures independently and, in 2 cases, involving left adrenal biopsy, beyond what was previously done by our unit.

In our practice, no strict protocol for systematic EBUS approach to mediastinum evaluation has been instituted. Whether procedure was based on systematic approach or targeted EBUS approach was dependent on individual bronchoscopists. For EUS-B FNA, systematic approach in accordance with EUSAT assessment tool is adhered to in training and practice. In a SCORE study [14], systematic combined approach yielded additional cases of diagnosed mediastinal nodal metastases and additional cases of change in nodal stage status. We will pursue further practice incorporating a protocolized combined systematic mediastinum assessment approach in future.

In conclusion, our experience suggests that there is relative ease in transition to combined EBUS EUS-B TBNA procedures for mediastinal lymphadenopathy and lung cancer diagnosis and staging for experienced bronchoscopist using a program which incorporates and extends EUS-B FNA training and supervision into daily real-life practice.

## Figures and Tables

**Figure 1 diagnostics-11-01021-f001:**
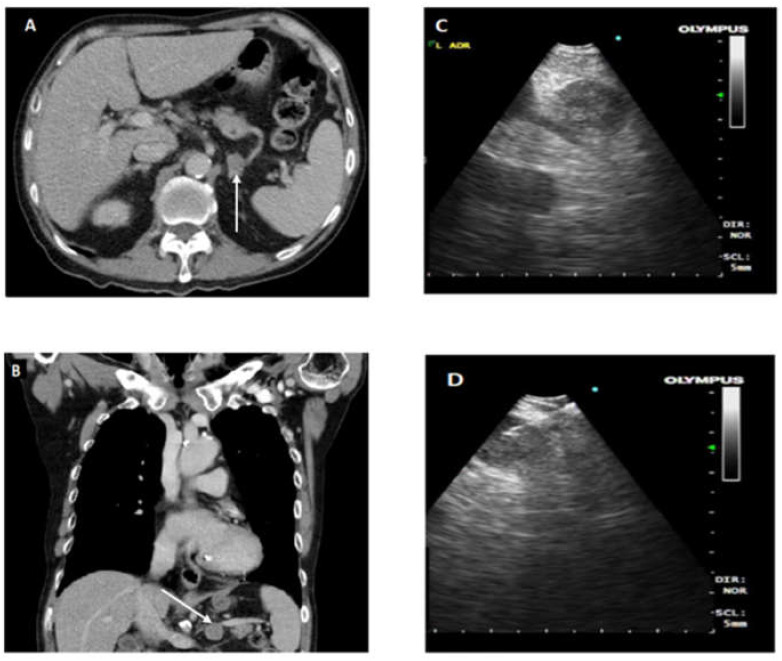
(**A,B**) CT axial and coronal sections, respectively. Arrowhead indicates enlarged left adrenal gland (LAG); (**C**) demonstrates EUS-B image of LAG; (**D**) illustrates EUS-B FNA of enlarged LAG with needle in target lesion.

**Figure 2 diagnostics-11-01021-f002:**
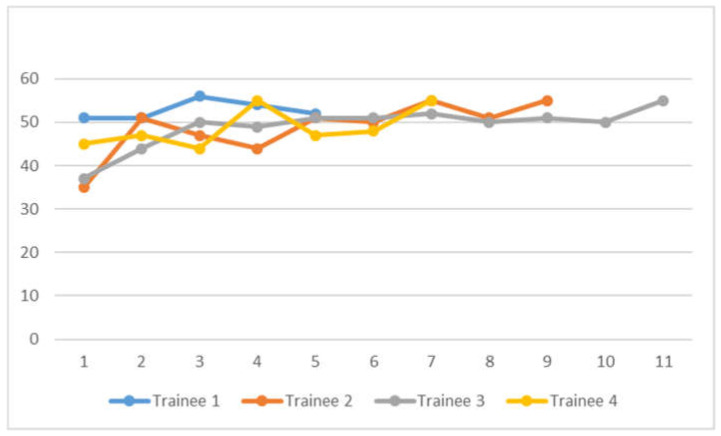
EUSAT Score plot of 4 Bronchoscopists performing EUS-B FNA.

**Table 1 diagnostics-11-01021-t001:** Characteristics of 32 patients who underwent combined EBUS EUS-B.

Patients	
Male	27
Female	5
Pre-Procedure diagnosis	
Malignancy	28
Tuberculosis	2
Sarcoidosis	2
Final Diagnosis	
Malignancy	23
Tuberculosis	2
Sarcoidosis	3
Benign	4
Number of Cases with 6 or more Months Follow up	27
Number of Cases with Further Tissue Biopsy	5
Number of Cases EUS-B FNA Cytology Report is Consistent with Final Diagnosis	28

**Table 2 diagnostics-11-01021-t002:** Demographics of experienced EBUS bronchoscopists participating in EUS-B FNA training.

Number of Bronchoscopists	4
Age (Mean)	43.5
Years Post Graduate	18.75
Years Post Specialization	10.5
Number of EBUS per Year	42.5

**Table 3 diagnostics-11-01021-t003:** Number of lymph nodes or lesions sampled according to stations by EBUS TBNA and EUS B FNA.

Lymph Node Station	EBUS TBNA	EUS B FNA
2R	1	0
4R	17	0
4L	7	18
7	8	22
8	0	1
11R	4	0
11L	3	0
LAG	0	2
Lung Mass	3	4
Average Size of LNs on CT (mm)		
Long Axis	21.5	20.8
Short Axis	16.2	14.3
Number of LNs Sampled per Patient	1.3	1.3
Number of Passes per Lymph Node Lesion	1.9	2.5

**Table 4 diagnostics-11-01021-t004:** Predictors of high EUSAT score (EUSAT > 50).

Predictors	Univariate OR (95% CI)	*p*-Value	Multivariate OR (95% CI)	*p*-Value
Short Axis Diameter (mm)	1.03 (0.97–1.08)	0.376	1.02 (0.96–1.08)	0.551
Non-4LE/7E Target Location	3.06 (0.53–17.7)	0.212	2.70 (0.45–16.3)	0.280

## Data Availability

Not applicable.

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
