# Peer review of "Transitioning to Combined EBUS EUS-B FNA for Experienced EBUS Bronchoscopist"

_diagnostics, 2021, doi:10.3390/diagnostics11061021_

Round 1

Reviewer 1 Report

The authors assessed the proficiency of the examiners in a relatively rare examination technique that combines EBUS with EUS-B FNA. They contend that if a respiratory physician who is proficient in EBUS did EBUS with EUS-B FNA, a few training cases will result in a high score for the assessor, and if they can maintain this level for more than three times, they could perform EUS-B FNA alone. I do not have an obvious opposition to the authors' statement. However, I would like to point out that there are some major problems with this manuscript.

  1. I believe it is especially important in this study to consider the location or size of the targeted lymph nodes, and whether the correct diagnosis was made in the examinations when the EUSAT score was low. I recommend reviewing the results from this perspective.

  1. Authors mentioned that there was no complication of mediastinitis or pneumothorax, but what about the oxygenation and hemostatic agents administered during the examination, and the degree of bleeding (and grading if possible)? Were these complications relevant to the EUSAT evaluation? I believe these safety evaluations are important to contend to have enough capacity to perform the examination alone.

  1. The IRB statement is wrote only as "pending". How is the IRB application process going? Although this research was not conducted beyond actual practice, I believe it is imperative to have an IRB evaluation in order to ensure ethical considerations.

  1. Due to the small number of participants in this study, it may be necessary to add a limitation that we cannot examine the correlation between the number of previous EBUS performed and the EUSAT score or the increase in EUSAT by repeat examinations.

  1. Please check and complete the Acknowledgments.

  1. Please correct the following words, if necessary.

                 specialisation                          specialization         (line 129)

                 para-oesophageal                  para-esophageal    (line 134)

Author Response

Reviewer 1

The authors assessed the proficiency of the examiners in a relatively rare examination technique that combines EBUS with EUS-B FNA. They contend that if a respiratory physician who is proficient in EBUS did EBUS with EUS-B FNA, a few training cases will result in a high score for the assessor, and if they can maintain this level for more than three times, they could perform EUS-B FNA alone. I do not have an obvious opposition to the authors' statement. However, I would like to point out that there are some major problems with this manuscript.

Thank you for your constructive and insightful comments. We have done all amendments on the manuscript in green. Please see attachment.

  1. I believe it is especially important in this study to consider the location or size of the targeted lymph nodes, and whether the correct diagnosis was made in the examinations when the EUSAT score was low. I recommend reviewing the results from this perspective.

Only 4 patients did not have correct diagnosis and multivariate regression cannot be done. Fortunately, we can show that short axis diameter and non-4LE/7E location were not associated with high EUSAT score. This would then allow a univariate analysis of high EUSAT score and correct diagnosis, as below.

Table. Predictors of high EUSAT score (EUSAT>50)

Predictors

Univariate OR

(95% CI)

P-value

Multivariate OR

(95% CI)

P-value

Short axis diameter (mm)

1.03 (0.97-1.08)

0.376

1.02 (0.96-1.08)

0.551

Non-4LE/7E target location

3.06 (0.53-17.7)

0.212

2.70 (0.45-16.3)

0.280

Using logistic regression, we did not find any association between short axis diameter and non-4LE/7E target location with high EUSAT score (>50), either on univariate or on multivariate analysis (Table). We therefore believe that the temporal improvement of EUSAT scores is attributable to a learning effect over time.

Association between correct diagnosis and high EUSAT score

In addition, there was no association between correct diagnosis and high EUSAT score (Fisher exact test, P = 0.109). This is likely due to the supervisor guiding the training bronchoscopists to sample lesions appropriately, even while the latter were learning.

Above statistical analysis and discussion are included in Results and Discussion section.

  1. Authors mentioned that there was no complication of mediastinitis or pneumothorax, but what about the oxygenation and hemostatic agents administered during the examination, and the degree of bleeding (and grading if possible)? Were these complications relevant to the EUSAT evaluation? I believe these safety evaluations are important to contend to have enough capacity to perform the examination alone.

In our audit of EUS B cases, there were no cases of hemostatic agents administered for bleeding. There were no cases with clinically significant bleeding.

Combined EBUS EUS B cases are routinely done with supplemental oxygen administered by nasal prongs up to maximum of 5 Litres per minute. None of EUS B case had desaturation requiring escalation of oxygen or mechanical ventilatory support.

  1. The IRB statement is wrote only as "pending". How is the IRB application process going? Although this research was not conducted beyond actual practice, I believe it is imperative to have an IRB evaluation in order to ensure ethical considerations.

We have applied for ethics. Currently awaiting approval. Application reference number 2021/00425

  1. Due to the small number of participants in this study, it may be necessary to add a limitation that we cannot examine the correlation between the number of previous EBUS performed and the EUSAT score or the increase in EUSAT by repeat examinations.

Thank you for suggestion. We have included the limitation in Discussion.

  1. Please check and complete the Acknowledgments.

“Much appreciation and gratitude to Professor Kim HJ and Doctors of Division of Pulmonary and Critical Care, Samsung Medical Centre, Seoul, South Korea for training and mentorship.”

  1. Please correct the following words, if necessary.

                 specialisation                          specialization         (line 129)

                 para-oesophageal                  para-esophageal    (line 134)

Thank you. We done the correction.

Reviewer 2 Report

This is an interesting and important report.

More detail on the method of performing EUS-B FNA would be useful rather than simply referring to previous publications  

This paper describes the value of the additional use of EBUS-B FNAB for diagnosis of lung cancer. This process has been described but the cytological processes involved have not. 4 experienced bronchoscopists were trained and their performance was assessed by "EUSAT" The EUSAT assessment process needs to be defined and the scoring process described. An estimate of the additional time required to perform the oesophageal procedure would be interesting! The performance scores of the operators as measured by EUSAT rapidly improved so that the additional procedure was not a problem for them. The text needs editing for simple grammatical correctness. Bill Musk

Author Response

This is an interesting and important report.

More detail on the method of performing EUS-B FNA would be useful rather than simply referring to previous publications  

This paper describes the value of the additional use of EBUS-B FNAB for diagnosis of lung cancer. This process has been described but the cytological processes involved have not. 4 experienced bronchoscopists were trained and their performance was assessed by "EUSAT" The EUSAT assessment process needs to be defined and the scoring process described. An estimate of the additional time required to perform the oesophageal procedure would be interesting! The performance scores of the operators as measured by EUSAT rapidly improved so that the additional procedure was not a problem for them. The text needs editing for simple grammatical correctness. Bill Musk

Thank you for your constructive comments and interest.

With regards to details on performing EUS-B FNA, I have CT and EUS-B image of left adrenal gland FNA which may require ethics board review and approval before publication. If there is interest and Editors are agreeable, I can work on including the images.

We will make a request for our cytology technician or pathologist colleague to work with us on more detailed description of cytological processes.

We have described EUSAT score as 12-item assessment method with scale 1-5. EUSAT assessment scoring was done on site or shortly after procedure. As supervisor may take over the procedure, hence some EUSAT assessments are done after procedure. 5 points was assigned for performance of itemized skill on first attempt. 4 points for performance at 2nd attempt with verbal guidance. 3 points for performance after more than 2 attempts. 2 points for performance after more than 2 attempts with verbal guidance. 1 point for inability to perform itemized skill and scope is taken over by supervisor.

In our practice, we attempt to perform all bronchoscopic diagnostic procedures in 1 session, our cases may involve, as an example EUS-B FNA followed by radial ultrasound transbronchial lung biopsy (TBLB) and or bronchoalveolar lavage (BAL). As an illustration, the documented procedural time of EUS-B FNA of left adrenal gland is 64 minutes and it involved TBLB at same session. A case of EUS-B FNA of 4L lymph node with no other bronchoscopic procedures done had documented procedure time of 27 minutes. The data I have on procedural timing is likely to overestimate due to additional procedures and time spend on direct supervision.

Round 2

Reviewer 1 Report

The authors have adequately answered my previous points in many of them. However, I would like to point out a major point and a minor point as well.

  1. Major point; The most serious problem is that the IRB has not yet approved the study at the time of this submission. I think it would be better to resubmit the manuscript after the IRB's approval and make necessary additions.
  2. Minor point; The author showed us that there were no complications associated with the examination, neither serious bleeding nor hypoxemia. While this indicates that EBUS with EUS-B FNA is a safe procedure for physicians with some proficiency, however, it may also indicate that the small number of patients in this study did not allow for adequate evaluation. In my opinion, this should be mentioned in the Discussion.

Author Response

  1. Major point; The most serious problem is that the IRB has not yet approved the study at the time of this submission. I think it would be better to resubmit the manuscript after the IRB's approval and make necessary additions.Thank you for your patience. IRB convenes every month. We will revert to Reviewers and Editors once approval is obtained. We understand that Journal deadline is in September 2021.
  2. Minor point; The author showed us that there were no complications associated with the examination, neither serious bleeding nor hypoxemia. While this indicates that EBUS with EUS-B FNA is a safe procedure for physicians with some proficiency, however, it may also indicate that the small number of patients in this study did not allow for adequate evaluation. In my opinion, this should be mentioned in the Discussion.Combined EBUS EUS-B-FNA has low complication rate in meta-analysis by Dhooria et al (Reference 6. Respiratory Care 2015;60;1040-1050) and Korevaar et al (Reference 7. Lancet Resp Med 2016:4:960-968). In Korevaar et al, serious adverse events were from EBUS TBNA, 7 adverse events out of 2171 patients. In more than 30 cases done at our center by author himself, audit did not reveal any complications of infection, bleeding, pneumothorax and hypoxaemia by EUS-B FNA. No training was performed for these cases, due mainly to scheduling. Training participants unable to be present for cases.For training bronchoscopist, our study advocates direct supervision and monitoring of competence with EUSAT assessment and score.In our center, safety culture is priority. Complications and adverse effects are reported swiftly to hospital authorities. Cessation of procedures to facilitate rigorous safety reviews are commonplace. Despite points above, I have no strong objections to including your suggestion into discussion.

Round 3

Reviewer 1 Report

The authors responded adequately. However, we need to wait for IRB approval before we publish this manuscript.